# Highly Dispersed Pt-Incorporated Mesoporous Fe_2_O_3_ for Low-Level Sensing of Formaldehyde Gas

**DOI:** 10.3390/nano13040659

**Published:** 2023-02-08

**Authors:** Seung Jin Jeon, Kyung Hee Oh, Youngbo Choi, Ji Chan Park, Hyung Ju Park

**Affiliations:** 1Electronics and Telecommunications Research Institute (ETRI), 218 Gajeong-ro, Yuseong-gu, Daejeon 34129, Republic of Korea; 2Department of Safety Engineering, Chungbuk National University, Cheongju 28644, Republic of Korea; 3Clean Fuel Research Laboratory, Korea Institute of Energy Research, Daejeon 34129, Republic of Korea; 4Department of Chemistry, Korea University, Seoul 02841, Republic of Korea; 5Energy Engineering, University of Science and Technology, Daejeon 34113, Republic of Korea

**Keywords:** formaldehyde gas sensor, Pt-incorporated mesoporous Fe_2_O_3_, high sensitivity

## Abstract

Highly dispersed Pt-incorporated mesoporous Fe_2_O_3_ (Pt/m-Fe_2_O_3_) of 4 μm size is prepared through a simple hydrothermal reaction and thermal decomposition procedures. Furthermore, the formaldehyde gas-sensing properties of Pt/m-Fe_2_O_3_ are investigated. Compared with our previous mesoporous Fe_2_O_3_-based gas sensors, a gas sensor based on 0.2% Pt/m-Fe_2_O_3_ shows improved gas response by over 90% in detecting low-level formaldehyde gas at 50 ppb concentration, an enhanced selectivity of formaldehyde gas, and a lower degradation of sensing performance in high-humidity environments. Additionally, the gas sensor exhibits similar properties as the previous sensor, such as operating temperature (275 °C) and long-term stability. The enhancement in formaldehyde gas-sensing performance is attributed to the attractive catalytic chemical sensitization of highly dispersed Pt nanoparticles in the mesoporous Fe_2_O_3_ microcube architecture.

## 1. Introduction

Among the various air pollutants, formaldehyde (HCHO) is a highly toxic substance which is commonly used in industries such as building construction and furniture [1,2]. Furthermore, due to its high toxicity, widespread distribution, and volatility, HCHO is a major cause of sick building syndrome (SBS). It is therefore classified as an indoor and outdoor air pollutant that poses a significant risk to human health [3,4], and the long-term exposure limit for HCHO has been set at a very low concentration. The Agency for Toxic Substances and Disease Registry (ATSDR) developed the minimal risk levels (MRLs) of HCHO at 40 ppb (1–14 days), 30 ppb (>14–364 days), and 8 ppb (≥365 days) [5]. Developing highly sensitive, stable, and selective gas sensors at these very-low-ppb-level concentrations of HCHO gas is thus crucial.

HCHO gas sensors based on metal oxide semiconductors are widely used because of their simple working principle, high sensitivity, portability, and low cost [6,7,8,9,10,11]. Additionally, a gas sensor based on p-type mesoporous metal oxide semiconductors, which conduct through positive-hole accumulation layers, provides a promising practical material platform for the low-level sensing of volatile organic compounds (VOCs), showing improved sensitivity, good selectivity, and lower humidity dependence [12,13,14,15,16].

Recently, well-designed metal oxide nanostructures with high surface area to volume ratios have been developed and applied to achieve effective gas sensing [17,18,19]. Furthermore, surface modification of the metal oxide structure by noble metals with unique electronic states and chemical properties has reduced the activation energy of adsorption and enhanced electron transfer efficiency, resulting in an improvement in gas-sensing performance [20,21,22]. The loading of appropriate Pt dopant in mesoporous semiconductor materials has been shown to improve the performance of gas sensors [23,24,25,26,27]. However, obtaining the optimal dopant ingredients and amounts while maintaining the consistent mesoporous structure with enhanced gas-sensing properties is challenging. Additionally, the incorporation of highly dispersed Pt nanoparticles (NPs) has been limited by the complex procedures involved with harmful chemicals and inhomogeneous dispersion [28,29,30,31,32].

In this study, we introduce a novel formaldehyde (HCHO) gas-sensing material consisting of Pt NPs and a mesoporous Fe_2_O_3_ (m-Fe_2_O_3_) structure, which is prepared by the thermal oxidation of cubic-shaped FeC_2_O_4_·2H_2_O particles as Fe_2_O_3_ precursors, and subsequently the additive thermal decomposition of Pt(acac)_2_ as a precursor of Pt NPs. The Pt-incorporated mesoporous Fe_2_O_3_ (Pt/m-Fe_2_O_3_) shows better sensitivity (Response: 2.13 at 50 ppb HCHO) than m-Fe_2_O_3_, excellent selectivity for formaldehyde gas, and a long-term stability of 120 days, even at high humidity.

## 2. Methods

### 2.1. Chemicals

Iron (III) nitrate nonahydrate (Fe(NO_3_)_3_·9H_2_O, ACS reagent, ≥98%), poly (vinyl pyrrolidone) (PVP, Mw = 55,000), D-(+)-glucose (ACS reagent, ≥99.5%), and platinum (II) acetylacetonate (Pt(C_5_H_7_O_2_)_2_, Pt(acac)_2_, 97%) were purchased from Aldrich (St. Louis, United States). All reagents were used as received without further purification.

### 2.2. Synthesis of Pt-Incorporated Mesoporous Fe_2_O_3_ (Pt/m-Fe_2_O_3_) Microcubes

For the synthesis of Pt/m-Fe_2_O_3_ particles, m-Fe_2_O_3_ powder was prepared with a simple thermal decomposition of the cubic-shaped FeC_2_O_4_·2H_2_O, which could be obtained with a hydrothermal reaction, as in a previous report [33]. FeC_2_O_4_·2H_2_O powders were thermally treated at 400 °C under a continuous flow of air (200 mL·min^−1^) to yield m-Fe_2_O_3_ powder. For the incorporation of Pt NPs, m-Fe_2_O_3_ (0.5 g) and Pt(acac)_2_ (2.0 mg) were homogeneously mixed by grinding the powders in a mortar for 5 min under ambient conditions. Subsequently, the mixture was transferred to an alumina boat in a tube-type furnace, and then heated with a ramp of 3.3 °C·min^−1^ to 400 °C. The final Pt/m-Fe_2_O_3_ powder was obtained by thermal treatment at 400 °C for 30 min under N_2_ flow (200 mL·min^−1^).

### 2.3. Characterization

Scanning electron microscope (SEM) images were obtained using a NovaNano SEM 450 (FEI, Hillsboro, OR, USA) by a Schottky field emission gun at 0.2~30 kV range with 1.0 nm ultimate resolution at 15 kV. A Talos F200X (Thermo Fisher Scientific, Waltham, MA, USA) with a 0.12 nm TEM information limit and a 0.16 nm STEM high-angle annular dark field (HAADF) resolution was used at 200 kV for transmission electron microscopy (TEM) analysis. Energy-dispersive X-ray spectroscopy (EDS) elemental mapping was performed using a high-efficiency detection system (Super X: 4 windowless SDD EDS system). The TEM samples were prepared by placing a few drops of the colloidal solutions on copper grids coated with Formvar carbon film (Ted Pella, Inc., Redding, CA, USA). Additionally, X-ray diffraction (XRD) patterns were measured on a Rigaku SmartLab X-ray diffractometer (Rigaku Corp., Tokyo, Japan) with Cu Kα radiation (λ = 1.54051 Å), with a scan speed of 3°·min^−1^ and a step size of 0.01° in 2*θ* range (20−80°). The N_2_-sorption isotherms were measured at −196 °C with a TriStar II 3020 surface area analyzer (Micromeritics Inc., Norcross, GA, UAS). Before measurement, the sample was degassed in a vacuum at 300 °C for 4 h. Raman spectra of the samples were collected using high-resolution Raman spectroscopy (LabRAM HR Evolution visible_NIR, HORIBA) with a 514 nm laser in the range of 100–3500 cm^−1^. The Pt contents of the samples were measured using ICP-OES (PerkinElmer AVIO500, Waltham, MA, USA). Thermogravimetry (TG) measurement was conducted using a SETARAM apparatus at a heating rate of 10 °C·min^−1^ from 40 to 800 °C under air flow (10 mL·min^−1^).

### 2.4. Gas-Sensing Measurement

The gas-sensing measurement system used was a laboratory-made system that could control working temperature, humidity level, and analyte gas concentration inside the reaction chamber. The fabricated gas sensor was located in the reaction chamber in the measurement system. Then, 20 ppm formaldehyde (RIGAS Co., Daejeon, Republic of Korea) gas was used as the analyte gas and diluted from 50 to 1000 ppb with dry air as the balance gas via mass flow controllers

Based on the requirements, humid air was supplied through a humidity chamber and temperature was controlled (by halogen lamp) in the chamber. To check the performance of the gas sensor, the electric resistance was measured and recorded with a digital system (Agilent 34970A, Culver, CA, USA). The performance of the gas sensor was primarily considered by its response value. The response (*R*) of a p-type semiconductor is calculated following Equation (1)
*R* = *R_gas_*/*R_air_*(1)
where *R_gas_* is the electric resistance when exposed to the target gas, and *R_air_* is the electric resistance in air conditions without the target gas.

## 3. Results and Discussion

### 3.1. Synthesis of Pt-Incorporated Mesoporous Fe_2_O_3_

The incorporation of Pt NPs was simply conducted by mixing the Fe_2_O_3_ powder and Pt(acac)_2_ followed by the thermal treatment of the mixture under an N_2_ flow condition at 400 °C (Figure 1). First, cubic-shaped FeC_2_O_4_·2H_2_O particles could be obtained as a precursor of m-Fe_2_O_3_ from a hydrothermal reaction of Fe(NO_3_)_3_·9H_2_O under the presence of PVP as a surfactant and glucose as a carbon source, as reported in previous work [33]. Additionally, the obtained FeC_2_O_4_·2H_2_O powder could be decomposed and further oxidized to m-Fe_2_O_3_ particles via thermal oxidation under air flow at 400 °C. The weight ratio of the Pt precursor (g)/m-Fe_2_O_3_ powder (g) was 0.004 in the mixture, and the incorporated wt% of Pt was 0.2 in the Pt/m-Fe_2_O_3_ sample. The Pt(acac)_2_ compound (m.p. = 249−252 °C) could be completely decomposed during thermal treatment at 400 °C, yielding highly dispersed Pt NPs deposited on the Fe_2_O_3_ surface.

The SEM image shows cubic-shaped Pt/m-Fe_2_O_3_ particles with crack-like large pores (Figure 1a). The average size of the Pt/m-Fe_2_O_3_ particles is observed to be 4.0 ± 0.8 μm by measuring the edges of 200 particles in the SEM images (Figure 1b). The shape and size of Pt/m-Fe_2_O_3_ are similar to pristine m-Fe_2_O_3_ (Appendix A). The TEM image shows the edge of the cubic Pt/m-Fe_2_O_3_ consisting of small particles (approximately 25–30 nm) and pores (Figure 1c). The high-angle annular dark field (HAADF)-TEM image shows the uniform and high dispersion of Pt particles (<2 nm sizes) on m-Fe_2_O_3_ (Figure 1d). The elemental mapping image of Pt (green color) and magnified TEM image demonstrate the uniform Pt distribution (Figure 1e,f). The high-resolution TEM (HR-TEM) image shows line spacings of 0.22 and 0.25 nm indexed to the Pt (111) and Fe_2_O_3_ (110) reflections, respectively (Figure 1g).

The X-ray diffraction (XRD) patterns in all Pt/m-Fe_2_O_3_ samples with the different Pt contents identically represent peaks at 2*θ* = 24, 33, and 36°, assigned to the reflections of the (012), (104), and (110) planes of a Fe_2_O_3_ rhombohedral structure (Figure 1h, JCPDS No. 33-0664). In the XRD spectra, no significant difference was observed between Pt/m-Fe_2_O_3_ and Pt-free m-Fe_2_O_3_ (Appendix A). Using the Debye–Scherrer equation based on the peak broadening of the (110) reflection in the Pt-doped samples, the Fe_2_O_3_ crystal domain sizes were measured to be around 35 nm, which matched well with the size observed in the TEM images. The N_2_ sorption experiment at −196 °C for the Pt/m-Fe_2_O_3_ exhibited type IV adsorption–desorption hysteresis (Figure 2a). The Brunauer–Emmett–Teller (BET) surface area and total pore volume were calculated to be 25 m^2^·g^−1^ and 0.13 cm^3^·g^−1^, respectively. With the Barrett–Joyner–Halenda (BJH) method, the pore size in the Pt/m-Fe_2_O_3_ particle was determined to be 21 nm in the desorption branch (Figure 2b). Because well-defined pores and small crystals can improve gas-sensing performance, we assumed that forming m-Fe_2_O_3_ microcubes with a long-range ordered framework consisting of suitable hematite crystals and nanopores with large surface areas could contribute to the significant response to VOCs. We consider that m-Fe_2_O_3_ microcubes with a large surface area, small grain size, and large contact area are sensitive and reliable gas-sensing materials. Furthermore, the incorporation of small noble metal nanoparticles may result in significant synergy for VOC detection.

In the thermogravimetric analysis (TGA), 0.2 wt% Pt/m-Fe_2_O_3_ showed a slight mass loss of 0.37 wt% in the range of 300 to 800 °C, demonstrating the presence of carbon residue in the sample (Figure 2c). Although this value is small, this showed a meaningful value similar to the ~0.24 wt% mass loss of p-type m-Fe_2_O_3_ published in the previous paper, which is different from the ~0 wt% mass loss of commercial n-type Fe_2_O_3_ nanoparticles [33]. We performed additional characterization experiments to confirm the carbon residue. However, any significant peaks for residual carbon were not detected in the Raman analysis, only showing the intrinsic peaks of α-Fe_2_O_3_ (Appendix A). A typical G band of carbon was not observed around 1560 cm^−1^ due to the very high intensity of Fe_2_O_3_ and a small amount of residual carbon. We believe that the presence of trace carbon impurities affecting the electric properties of Pt/m-Fe_2_O_3_ could be indirectly confirmed by TGA.

With inductively coupled plasma optical emission spectrometry (ICP-OES) measurements, the Pt contents in the nominal Pt/m-Fe_2_O_3_ samples were obtained to be 0.14 wt% at 0.1 wt% Pt/m-Fe_2_O_3_, 0.23 at 0.2 wt% Pt/m-Fe_2_O_3_, and 0.43 wt% at 0.4 wt% Pt/m-Fe_2_O_3_, respectively.

### 3.2. Performance of HCHO Gas Sensing

The 0.2% Pt/m-Fe_2_O_3_ and m-Fe_2_O_3_ sensing materials were prepared in a paste form by mixing them with glues, after which they were screen-printed onto interdigitated electrodes. After thermal treatment at 400 °C for 2 h, the fabricated gas sensors were investigated by an electrical resistance measurement system. The formaldehyde gas (20 ppm) samples were mixed with dry air to meet the desired analyte concentration, which was in the range of 50–1000 ppb, using mass flow controllers.

The working temperature has a significant influence on gas-sensing properties. As a well-known property of semiconductors, the resistance of Pt/m-Fe_2_O_3_ and m-Fe_2_O_3_ showed a gradual decrease with temperature increase. To determine the optimal concentration of Pt incorporation and suitable working temperature for the gas-sensing response, the pure m-Fe_2_O_3_ and several Pt/m-Fe_2_O_3_ samples (0.1, 0.2, and 0.4 wt% Pt-incorporated) were measured at different temperatures from 260 to 300 °C for 100 ppb HCHO gas. The response shape peaked at 275 °C, and 0.2% Pt/m-Fe_2_O_3_ was the most sensitive response, as shown in Figure 3. When the Pt content was higher than the optimal addition, the response was poor again due to the agglomeration of Pt [31]. Therefore, 0.2% Pt/m-Fe_2_O_3_ was selected, and 275 °C was decided as the working temperature for the investigations of the HCHO gas-sensing performance. Additionally, the HCHO gas response of 0.2% Pt/m-Fe_2_O_3_ was significantly higher than that of m-Fe_2_O_3_ at 275 °C.

Figure 4a shows the dynamic resistance change curves of 0.2% Pt/m-Fe_2_O_3_ and m-Fe_2_O_3_ to the HCHO gas in the range of 50–1000 ppb at 275 °C. In the initial air, the base resistance of 0.2% Pt/m-Fe_2_O_3_ was lower than that of m-Fe_2_O_3_ due to the electronic transition change after the addition of Pt [28]. Moreover, the 0.2% Pt/m-Fe_2_O_3_ had a larger resistance change ratio (response) than m-Fe_2_O_3_, while the target HCHO gas was injected (Figure 4b). Unlike other general Fe_2_O_3_ nanoparticles, our iron oxides (the Pt/m-Fe_2_O_3_ and m-Fe_2_O_3_) showed typical p-type electrical properties in which the resistance increased upon exposure to the HCHO gas. However, the conduction type of iron oxide was occasionally changed from n- to p-type when high temperatures were applied or impurities were added [34,35]. In a previous study, we discovered that exposing m-Fe_2_O_3_ to HCHO gas increased its resistance [33]. The presence of carbon impurities, which was the major cause of the p-type properties of the Pt/m-Fe_2_O_3_, can be expected by a mass loss of 0.2% Pt/m-Fe_2_O_3_ in the TGA (Figure 2c). Despite heat treatment at 400 °C, an extremely small amount of carbon residue remained in 0.2% Pt/m-Fe_2_O_3_, primarily originating from the used PVP. Furthermore, the p-type properties of the gas sensors could be determined. We think that this was also shown by the more dynamic change obtained in terms of resistance with the increasing concentration of HCHO gas, whereby the response was better with a higher concentration of the target gas. Based on the resistance change, the response was calculated, as shown in Figure 4b. Additionally, the constant response for three consecutive measurements at 1000 ppb HCHO gas indicated the good repeatability of the gas sensing. The fitted line in Figure 4c depicts the relation between the response and concentration of HCHO gas at 275 °C. The responses of 0.2% Pt/m-Fe_2_O_3_ were 2.13, 2.60, 3.37, 3.96, 4.43, 5.19, 5.64, and 5.93 at 50, 100, 200, 300, 400, 600, 800, and 1000 ppb HCHO gas, respectively. We observed that the response of the 0.2% Pt/m-Fe_2_O_3_ was higher than that of the pristine m-Fe_2_O_3_ with a good linear shape. Particularly, response 2.13 at the low concentration of 50 ppb HCHO gas was very significant. The limit of detection (LOD) could be calculated following Equation (2) [36]:LOD = 3σ/s(2)
where σ is the standard deviation and s is the slope of the calibration curve. The result of the calculated LOD was 7.38 ppb at 0.2% Pt/m-Fe_2_O_3_ and 12.27 ppb at m-Fe_2_O_3_, implying that the 0.2% Pt/m-Fe_2_O_3_ gas sensor is more sensitive than the m-Fe_2_O_3_ sensor at extremely low concentration levels of HCHO gas. Details of the n- and p-type metal oxide semiconductor gas sensors are shown in Table 1.

Developing a good gas sensor is required to maintain high responses in high-humidity environments. Generally, metal oxide semiconductor gas sensors become less sensitive in such environments because water molecules adsorb on the sensing material surface rather than oxygen, and then the base resistance value increases [47,48] (Appendix A). The measurement in the humid environment was set at 200 ppb HCHO gas at 300 °C, rather than 275 °C, due to base resistance instability at high-humidity conditions. At 89% relative humidity (RH), the HCHO sensitivity performance of pristine m-Fe_2_O_3_ dropped to 48% compared to the dry air conditions. On the other hand, that of 0.2% Pt/m-Fe_2_O_3_ was only 25% lower in the same conditions. Even under the condition of 56% RH, similar properties of sensor performance are shown depending on changes in sensing materials, as in the case of 89% RH. The HCHO sensitivity performance of pristine m-Fe_2_O_3_ dropped to 35%, while that of 0.2% Pt/m-Fe_2_O_3_ decreased by only 18%, as shown in Figure 5a. Although high humidity unfortunately affects the sensitivity performance degradation of the gas sensor, the HCHO sensitivity performance of 0.2% Pt/m-Fe_2_O_3_ improved to nearly twice that of m-Fe_2_O_3_ at high-humidity conditions.

To determine the selectivity, several gases (HCHO, CH_3_COCH_3_, C_2_H_5_OH, C_6_H_6_, NO_2_, SO_2_, and NH_3_) were tested at the same concentration at 500 ppb, except CO_2_, which was tested at 500 ppm, and at the same temperature of 275 °C. Figure 5b indicates that 0.2% Pt/m-Fe_2_O_3_ and m-Fe_2_O_3_ showed good selectivity with HCHO gas. Additionally, the response ratio of HCHO against other gases (R_HCHO_/R_another gas_) of 0.2% Pt/m-Fe_2_O_3_ ranged from 1.68 to 4.50, whereas pristine m-Fe_2_O_3_ against other gases (R_HCHO_/R_another gas_) ranged from 1.25 to 3.26. The results show that the HCHO gas selectivity of 0.2% Pt/m-Fe_2_O_3_ is better than that of m-Fe_2_O_3._

Practical applications require the long-term stability of gas sensors. To confirm the long-term stability of the 0.2% Pt/m-Fe_2_O_3_ gas sensor, it was consecutively measured with base resistance in air conditions for 120 h. The average resistance was 10.91 MΩ and standard deviation was 0.995. The response value at initial measurement was compared with responses after 90 and 120 days towards 500 ppb HCHO gas at 275 °C. The HCHO gas response was well maintained during this period (4.93 at initial test, 4.84 at 90 days, and 4.82 at 120 days, respectively), as shown in Appendix A. The long-term stability of the HCHO gas sensor, while having a high response to HCHO gas, was verified for 120 days. Additionally, the response and recovery times of 0.2% Pt/m-Fe_2_O_3_ to 500 ppb HCHO at 275 °C were measured to be 22 and 431 s, which was similar to m-Fe_2_O_3_ (63 and 395 s in previous results [33]).

### 3.3. Sensing Mechanism

When p-type semiconductors such as the Pt/m-Fe_2_O_3_ are exposed to ambient air, oxygen species are absorbed and ionized on the surface area of the sensing materials by capturing electrons (e^−^) and generating holes (h^+^) in the band of the sensing materials as [8]
O_2(ads)_ + e^−^ → O_2(ads)_^−^ (80–150 °C)(3)
O_2(ads)_^−^ + e^−^ → 2O_(ads)_^−^ (150–260 °C)(4)
1/2O_2(ads)_^−^ + e^−^ → O_(ads)_^2−^ (300–500 °C)(5)

If the reducing gases (such as HCHO) are in contact with the sensing materials, the ionized oxygen releases the electrons, and, subsequently, the released electrons are injected into the hole-accumulation area and recombine with holes as
HCHO_(ads)_ + 2O_(ads)_^−^ → CO_2(gas)_ + H_2_O_(gas)_ + 2e^−^(6)

Consequentially, the change in the hole accumulation layer due to the reaction with the reducing gases leads to a resistance change because the charge carriers mainly move through the holes at the surface of the sensing materials.

Figure 6 depicts the sensing mechanism through the transition of the hole-accumulation layer of each sensing material (Pt/m-Fe_2_O_3_ and m-Fe_2_O_3_), as well as the contribution of Pt nanoparticles to enhance the sensitivity of HCHO gas sensing. Several studies have discussed the decisive role of noble metals as a catalyst because absorbed oxygen species in Pt nanoparticles could be ionized more easily, leading to a spill-over effect, while the surfaces of sensing materials respond to oxygen molecules [23,49,50]. The incorporated Pt nanoparticles might serve to increase the absorption and ionization of oxygen molecules by lowering the activation energy of the sensing material [37,40,51]. Figure 6a shows that Pt/m-Fe_2_O_3_ acquires a wider hole-accumulation layer than the pristine m-Fe_2_O_3_ in air conditions. The surface area of Pt/m-Fe_2_O_3_ has more ionized oxygen that could be reacted with the HCHO gas than the pristine m-Fe_2_O_3_. Figure 6b displays the narrowed hole-accumulation layer when the surface of the sensing material is in contact with the HCHO gas, and the degree of the reduction in the hole accumulation layer of Pt/m-Fe_2_O_3_ is bigger than that of m-Fe_2_O_3_. The sensitivity of the semiconductor gas sensors was decided at the scale of the resistance change, and depends on the extent of the surface reaction between the sensing materials and the target gas. Therefore, Pt-incorporated nanoparticles in m-Fe_2_O_3_ play a critical role as enhancers.

## 4. Conclusions

In this study, highly dispersed Pt-incorporated mesoporous Fe_2_O_3_ microcubes were synthesized as a p-type by the facile thermal treatment of mixture powders such as m-Fe_2_O_3_ and Pt precursor without harmful chemicals and inhomogeneous dispersion. Furthermore, electric studies of 0.2% Pt/m-Fe_2_O_3_ were conducted to confirm the properties of the HCHO gas sensor. The sensor had a higher sensitivity (R = 2.13 at 50 ppb HCHO), high selectivity to formaldehyde gas, high performance under high humidity conditions, and similar long-term stability (retention time of 120 days) compared with a previous m-Fe_2_O_3_ gas sensor. Particularly, the response of 0.2% Pt/m-Fe_2_O_3_ at the low-level formaldehyde gas concentration (50 ppb) improved by over 90% compared with our previous m-Fe_2_O_3_ because of the spill-over effect by the Pt NPs. The sensitivity performance of 0.2% Pt/m-Fe_2_O_3_ was almost twice as high as that of m-Fe_2_O_3_ in high-humidity conditions. This means that 0.2% Pt/m-Fe_2_O_3_ could be a good potential sensing material for HCHO gas for detecting very low ppb levels.

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
