# Peer review of "Highly Dispersed Pt-Incorporated Mesoporous Fe2O3 for Low-Level Sensing of Formaldehyde Gas"

_nanomaterials, 2023, doi:10.3390/nano13040659_

Round 1

Reviewer 1 Report

The paper is very good written, all experiments are suitable and supported by explanations. I suggest to add more information about spill-over effect as one of main mechanisms for enhancing the sensing properties.

Reviewer 2 Report

The manuscript describes brick-shaped Fe2O3 microparticles incorporated with Pt nanoparticles for formaldehyde gas sensing. The results are consistent and are presented in a logical manner. The manuscript is of an interest for the readership of the journal. Before the further consideration of the manuscript for the possible publication in Nanomaterials, the following concerns must be addressed:

1. There is no evidence for the statement, that the weight loss in TG curve (Fig. S3) corresponds to carbon impurity. TG-MS, Raman and/or IR spectroscopy can help to identify the exact composition of the sample.

2. As authors claimed, the temperature of 400oC was not enough for the FeC2O4 decomposition and complete carbon (?) release (P. 7; Figure S3). Why was this temperature used for the synthesis, not higher?

3. TG data are presented from 300oC, while in characterization section, the thermal analysis is said to start from room temperature. Why the TG plot was cut?

4. Conclusion are non-specific, too short and too general. Please make the conclusion section more focused.

5. The reaction mechanism is not clear from the equations and the figure. The mechanism of hole generation is not presented.

6. How did the authors detect, that circle areas in Fig. 1f correspond to Pt NPs? The same question is for Fig. 1g.

7. The characterization (ISP AES, TG, XRD) of samples with 0.1 and 0.4 wt% of Pt is not presented, even in Supplementary materials. Why these data were omitted?

8. The Characterization section contains no information on the instrument parameters, such as accelerating voltage, working distance, magnification, detectors used for SEM and TEM; diffraction angles range, step and accumulation time for XRD, etc.

To summarize, I recommend a minor revision.

Reviewer 3 Report

This work presents an interesting material that can be used for gas sensing; however, some issues could be improved.

Please point out how the conductivity type was estimated. The authors mention p-type, but no data are presented.

The adsorption/desorption curves should be added to the main text and the TEM images.

The conclusions can be broadened by describing the most significant results. In this form, they rather look like highlights.

The formatting is incorrect.
